# Pulsed axial epitaxy of colloidal quantum dots in nanowires enables facet-selective passivation

Yi Li[1,2], Tao-Tao Zhuang[1,3], Fengjia Fan[4,5], Oleksandr Voznyy [3], Mikhail Askerka[3], Haiming Zhu[6], Liang Wu[1], Guo-Qiang Liu[1], Yun-Xiang Pan[7], Edward H. Sargent [3] & Shu-Hong Yu [1,2]

Epitaxially stacking colloidal quantum dots in nanowires offers a route to selective passivation of defective facets while simultaneously enabling charge transfer to molecular adsorbates – features that must be combined to achieve high-efficiency photocatalysts. This requires dynamical switching of precursors to grow, alternatingly, the quantum dots and nanowires – something not readily implemented in conventional flask-based solution chemistry. Here we report pulsed axial epitaxy, a growth mode that enables the stacking of multiple CdS quantum dots in ZnS nanowires. The approach relies on the energy difference of incorporating these semiconductor atoms into the host catalyst, which determines the nucleation sequence at the catalyst-nanowire interface. This flexible synthetic strategy allows precise modulation of quantum dot size, number, spacing, and crystal phase. The facet-selective passivation of quantum dots in nanowires opens a pathway to photocatalyst engineering: we report photocatalysts that exhibit an order-of-magnitude higher photocatalytic hydrogen evolution rates than do plain CdS quantum dots.

[1] Division of Nanomaterials & Chemistry, Hefei National Research Center for Physical Sciences at the Microscale, CAS Center for Excellence in Nanoscience, Hefei Science Center of CAS, Collaborative Innovation Center of Suzhou Nano Science and Technology, Department of Chemistry, University of Science and Technology of China, Hefei 230026, China. [2] Anhui Key Laboratory of Condensed Matter Physics at Extreme Conditions, High Magnetic Field Laboratory, Chinese Academy of Sciences, Hefei 230031, China. [3] Department of Electrical and Computer Engineering, University of Toronto, 35 St George Street, Toronto, Ontario M5S 1A4, Canada. [4] CAS Key Laboratory of Microscale Magnetic Resonance and Department of Modern Physics, University of Science and Technology of China, Hefei, Anhui 230026, China. [5] Synergetic Innovation Center of Quantum Information and Quantum Physics, University of Science and Technology of China, Hefei, Anhui 230026, China. [6] Department of Chemistry, Zhejiang University, Hangzhou, Zhejiang 310027, China. [7] School of Chemistry and Chemical Engineering, Hefei University of Technology, Hefei, Anhui 230009, China. These authors contributed equally: Yi Li, Tao-Tao Zhuang. Correspondence and requests for materials should be addressed to E.H.S. (email: ted.sargent@utoronto.ca) or to S.-H.Y. (email: shyu@ustc.edu.cn)

Thanks to their size-tailored bandstructures, colloidal quantum dots (QDs) represent a class of materials of interest in nanoelectronics and nanophotonics[1–6]. The trap states in colloidal quantum dots, which typically arise from surface dangling bonds, can strongly localize photoexcited carriers and impede charge carrier extraction[7,8]. Surface-coordinated ligands such as phosphonic acids, carboxylic acids, and halides mitigate this problem[9–11]; however, efficient passivation for chalcogen-rich (111) facets is yet to be developed, largely due to the fact that most ligands are Lewis bases that do not bind to electron-rich dangling bonds[8,9].

Epitaxial shell passivation offers an alternative route to decoupling surface traps from the core material; however, the resulting charge carrier confinement inside the cores limits applications that require carrier transfer and extraction[12,13], such as photocatalysis. Technologies that seek simultaneously to achieve surface passivation and charge transfer in quantum dots remain an important goal.

Recently, single and multiple QDs have been axially stacked in a nanowire (i.e. quantum-dots-in-nanowire, denoted as QDNW) via gas-phase substrate-based methods[14–17], such as selective-area epitaxy and vapor-liquid-solid (VLS) growth. These have been used to showcase promising applications as single-photon sources[18–21] and lasers[22–24]. Intriguingly, for semiconductors with zinc blende (ZB) and wurtzite (WZ) structures, the epitaxial direction of QDNWs tends to be $[111]_{ZB}/[0001]_{WZ}$.

We hypothesized that trap state recombination on the (111) facets of QDs could be mitigated, yet carrier extraction from the sidewalls maintained, in the QDNW system, thereby enhancing photocatalysis. However, these substrate-based methods produce thick nanowires with diameters of tens to hundreds of nanometers. Given that the large surface-to-volume ratio plays a key role in photocatalysts, and wide bandgap tunability offers new avenues for performance optimization, we sought strategies that would permit us to reduce the size of QDNW architectures[25].

Colloidal synthesis provides access to small-sized QDs and nanowires[26,27]. Unfortunately, despite its tremendous development in the past decades[28–33], controllable axial epitaxy of QDs within nanowires remains technologically challenging, especially via one-pot growth approach. Although cation exchange reactions have been used to prepare colloidal superlattices exhibiting high stability[34], a high–level control over structural parameters has yet to be demonstrated – a consequence of fast reaction kinetics. Solution-liquid-solid (SLS) and solution-solid-solid (SSS) methods have been used to grow axial heteronanowires[35]. In particular, the in-situ growth of axial heteronanowires on a substrate has been achieved via a flow-based SLS method[36] – chemical precursors inside the microfluidic reactor were dynamically switched to grow the distinct segments. However, in colloidal syntheses nanowires are instead suspended in solution, and precursors are not readily removed as in substrate-based methods, a fact that restricts dynamic control over precursor concentrations and resulting composition along the nanowire[37,38].

Here we show that, in a catalyst-assisted growth process, the growth of semiconductor nanowires can be switched by injecting precursors for quantum dots, which are then incorporated into the host catalyst and nucleate first at the catalyst-nanowire interface. Once the injected precursors are depleted, the original semiconductor will resume its growth. Thus, we can achieve the dynamical switching of precursors in conventional flasks. We term this one-pot synthetic strategy pulsed axial epitaxial growth (PAEG), highlighting its ability to modulate the stacking of multiple CdS QDs in ZnS nanowires by controlling the pulsed introduction of the Cd precursor. These selectively-passivated architectures provide a platform to achieve superior photocatalytic performance in cocatalyst-free CdS-based photocatalysts.

## Results

**Design principles of QDNWs.** In the SSS-based growth strategy (Fig. 1a), a solution-phase semiconductor precursor is incorporated into a solid superionic conductor catalyst to form solid solution through high-density cation vacancies[39]. When the concentration of a foreign cation in the catalyst reaches supersaturation, the semiconductor starts to grow at the solid-solid interface in a layer-by-layer manner. SLS growth is similar to SSS growth, except that liquid nanoparticles are used instead as the catalyst.

In order to fabricate QDNWs with a finely controlled interface and QD composition, size, and spacing, it is important to use a slow growth rate. According to the Gibbs-Thomson (GT) effect[36,40], higher supersaturation of foreign atoms in catalysts gives rise to a higher growth rate. For this reason, SSS growth rather than SLS growth is preferred because a lower solubility and supersaturation in the solid catalyst can be achieved, thereby allowing for slower growth[39,41]. The low solubility facilitates faster depletion of foreign atoms in the solid catalyst, resulting in a sharper heterointerface. Aggregation or coalescence of the small catalyst, which takes place in the SLS growth, can be mitigated in SSS growth[42,43].

Here we selected – in order to stack axially an array of spaced CdS QDs in ZnS nanowires in a one-pot growth – solid $Ag_2S$ as the host catalyst. With the coexistence of Zn and Cd precursors in the flask, the incorporation sequence of foreign semiconductors into catalysts is key to determine the corresponding compositional distribution in the nanowires.

The following reactions describe single Cd/Zn atom incorporation:

$$Ag_{2n}S_n + CdR_2 \rightarrow Ag_{2n-1}S_nCd + \text{byproducts} \qquad (1)$$

$$Ag_{2n}S_n + ZnR_2 \rightarrow Ag_{2n-1}S_nZn + \text{byproducts} \qquad (2)$$

where $CdR_2$ and $ZnR_2$ represent the Cd (Cadmium diethyldithiocarbamate) and Zn (Zinc diethyldithiocarbamate) precursors, respectively. In each reaction, the resultant catalysts ($Ag_{2n-1}S_nCd$ and $Ag_{2n-1}S_nZn$) can be either neutral (+0) or singly-charged (+1). Thus, we define the relative intercalation energy (RIE) of Cd/Zn atoms into $Ag_2S$ lattice as:

$$RIE = E[Ag_{2n-1}S_nCd] + E[ZnR_2] - E[Ag_{2n-1}S_nZn] - E[CdR_2] \qquad (3)$$

The relative intercalation energies for Cd/Zn atom incorporating into the $Ag_2S$ lattice were calculated using density functional theory (DFT) (Fig. 1b and Supplementary Fig. 1). For the neutral insertion, RIE is −0.53 eV per unit cell, indicating that Cd is energetically preferred over Zn for incorporation into $Ag_2S$. A similar result was also obtained for the charged insertion with RIE of −0.29 eV per unit cell. Therefore, once the Cd precursor is added, the growth of ZnS nanowires switches to epitaxial growth of CdS QDs, and re-epitaxy of ZnS can take place after the depletion of Cd precursor (Fig. 1c). Thus, we dynamically switch the axial composition in nanowires by controlling the introduction of the Cd precursor, an advantage not readily achieved in prior flask-based syntheses.

**Controlled synthesis of colloidal CdS–ZnS QDNWs.** Figure 2a and Supplementary Figs. 2, 3 present low-magnification transmission electron microscopy (TEM) images of $m$CdS-$n$ZnS QDNWs, where $m$ is the number of CdS QDs (also addition times of Cd precursor) and $n$ is the number of ZnS segments. With increasing times of Cd addition, we observed an increasing number of CdS QDs in the nanowire, as seen in the contrast in

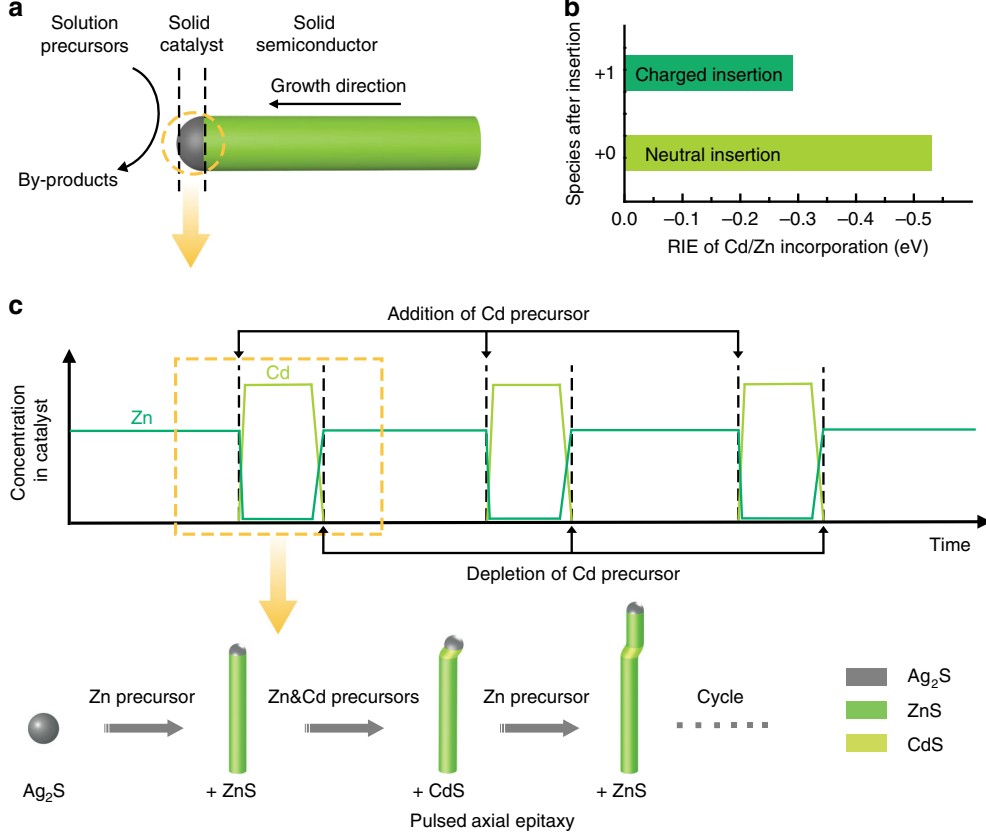

**Fig. 1** One-pot growth of colloidal QDNWs via pulsed axial epitaxy. **a** Schematic of the SSS growth mechanism for one-dimensional (1D) semiconductor nanowires. **b** DFT-calculated relative intercalation energies (RIE) for neutral (+0) and charged (+1) insertion of Cd/Zn into Ag$_2$S catalysts, respectively. In both cases, Cd insertion is energetically favorable over Zn. **c** (Upper panel) Schematic concentration evolutions of semiconductor atoms in catalysts with reaction time during the pulsed axial epitaxy. Addition of Cd precursor results in overwhelming incorporation of Cd into the Ag$_2$S host catalysts compared with that of Zn. In each cycle, CdS QDs epitaxially stack in nanowires right after the introduction of Cd precursor, followed by re-growth of ZnS segments, enabling the in-situ axial epitaxy of QDs in nanowires (bottom panel)

the TEM images. Correspondingly, the UV-vis absorption spectra and powder X-ray diffraction (PXRD) patterns (Supplementary Fig. 4) suggest that the number of CdS QDs in each nanowire increases stepwise as well. Without replenishing Zn precursor during the growth, we stacked up to five CdS QDs in a single nanowire (Supplementary Fig. 5).

We further used EDS element mapping (Fig. 2b and Supplementary Fig. 6) and linescans (Fig. 2c and Supplementary Fig. 7) to characterize the axial composition distributions in representative 3CdS-4ZnS QDNWs. We found that CdS QDs are incorporated into ZnS nanowires with substantially equal spacing, a finding that is further confirmed using dark-field scanning TEM (STEM) images (Fig. 2d–f and Supplementary Fig. 8). The typical high-resolution TEM (HRTEM) images (Fig. 2g and Supplementary Fig. 9) of ZnS/CdS/ZnS segments reveal two epitaxial modes along the [111]$_{ZB}$/[0001]$_{WZ}$ crystalline direction: ZB-ZnS/ZB-CdS/WZ-ZnS epitaxy and WZ-ZnS/ZB-CdS/WZ-ZnS epitaxy (see details in Supplementary Note 1 and Supplementary Fig. 10). Fast Fourier transforms (FFTs) of the HRTEM images confirm the high crystallinity of CdS segments and a lack of observable stacking faults. We interpret these images as consistent with the view that strain induced by a lattice mismatch is elastically released by the lateral surface[44,45]. The axial length of CdS segments is 3.9 ± 0.5 nm (Fig. 2h), which is close to 12 monolayers of CdS (111) facet, in agreement with the view that CdS segments in our QDNWs are within the quantum-confined size regime (exciton Bohr diameter of CdS is ca. 5.5 nm)[46].

We sought to further challenge the flexibility of the PAEG strategy, and found that – with controlled growth temperature, time, and introduction sequences of precursors – we modulate controllably the size of each segment in the QDNWs. The size of initially-nucleated Ag$_2$S catalysts increases when we raise the temperature from 190 °C to 250 °C, yielding nanowires with larger diameter (from 4.6 nm to 12 nm, Fig. 3a, b, Supplementary Fig. 11, and Supplementary Note 2).

Using time-dependent growth experiments, we find that QDNW growth proceeds at a constant rate (roughly 1 nm min$^{-1}$, see Fig. 3c and Supplementary Fig. 12). This suggests that variation in the concentration of the Zn precursor in the early stages has a modest influence on the nanowire growth rate. We can replenish the Zn precursor and elongate ZnS segments that sandwich the CdS QDs (Supplementary Figs. 13, 14). In this way we control the spacing of CdS QDs (Fig. 3d and Supplementary Fig. 15) via growth time. As a proof of concept, we further synthesized ultralong QDNWs with 10 CdS QDs (i.e., 21 segments in total, see Fig. 3e and Supplementary Fig. 16). We also demonstrated that we can increase the length of CdS segments by repeatedly injecting the Cd precursor (Supplementary Fig. 17).

**Photocatalytic performance and physical mechanism analysis.** We explored the application of the QDNW architecture in photocatalytic water splitting. In the as-synthesized QDNWs, CdS (111) facets are selectively passivated by ZnS segments, allowing long-lived charges from the sidewalls of CdS segments (Fig. 4a) to

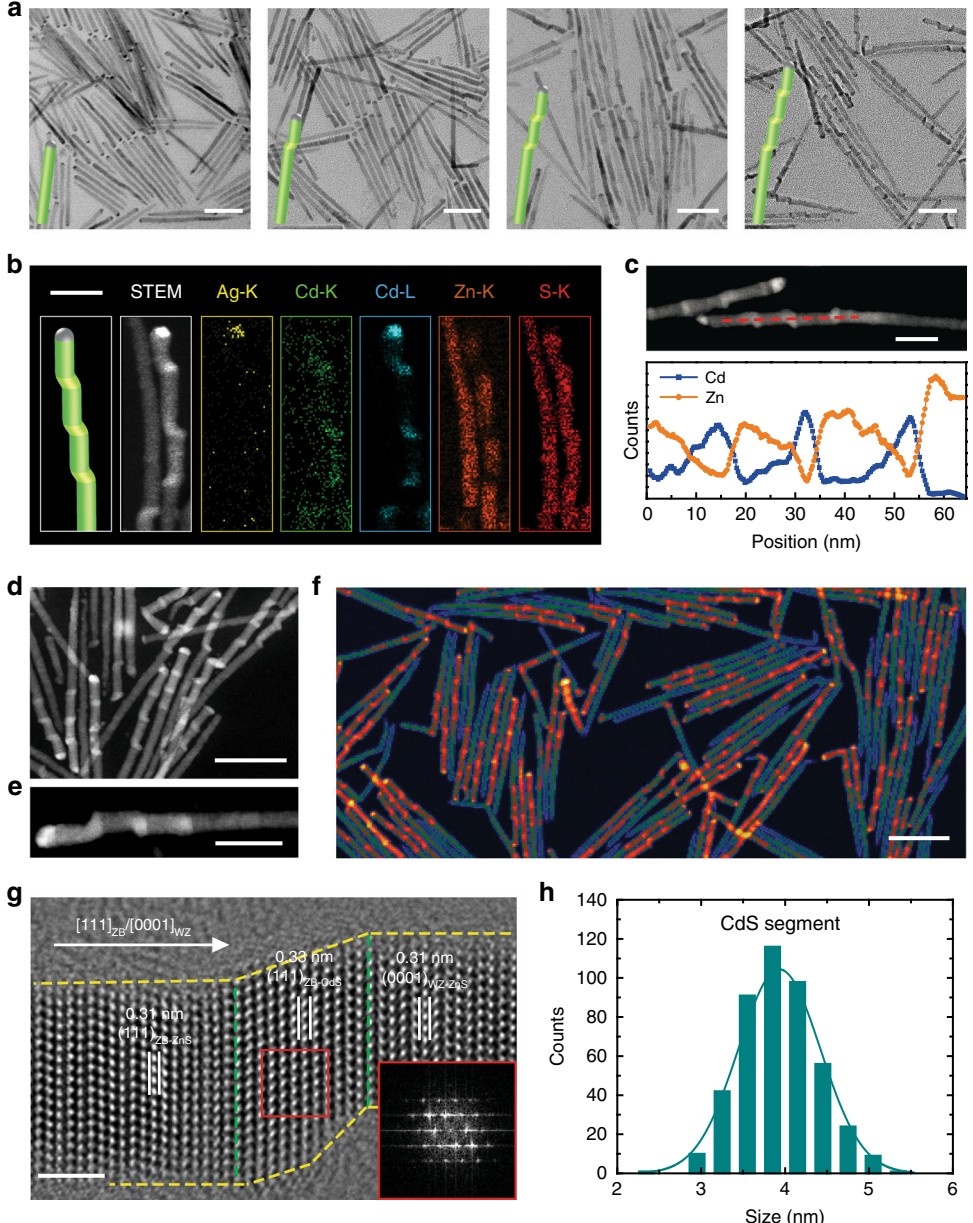

**Fig. 2** Synthesis and characterization of CdS–ZnS QDNWs. **a** TEM images of QDNWs with zero to three CdS QDs (from left to right). **b** Dark-field STEM images and EDS element mapping images of 3CdS-4ZnS QDNWs, shown with Ag-K (yellow), Cd-K (green), Cd-L (blue), Zn-K (orange), and S-K (red) signals. **c** EDS line-scan profiles of the 3CdS-4ZnS QDNW along the red dash line in dark-field STEM image. **d** Representative dark-field STEM image of 3CdS-4ZnS QDNWs. **e** Enlarged dark-field STEM image of single 3CdS-4ZnS QDNW. **f** Low-magnification pseudocolored dark-field STEM image of 3CdS-4ZnS QDNWs. CdS QDs with high contrast are uniformly distributed in nanowires. **g** Representative HRTEM image of one CdS QD sandwiched between two ZnS segments, featuring a $[111]_{ZB}/[0001]_{WZ}$ epitaxial direction. Inset: the FFT image of CdS segment (red square). **h** Axial length distribution of CdS QDs in 3CdS-4ZnS QDNWs. Scale bars are 50 nm for **a**, **d**, and **f**, 20 nm for **b**, **c**, and **e**, and 2 nm for **g**, respectively

participate in the redox process. We further investigated the hydrogen evolution rates of different photocatalysts (Fig. 4b and Supplementary Fig. 18) under visible-light irradiation (> 420 nm). The 3CdS-4ZnS QDNWs exhibit an enhanced hydrogen evolution rate, achieving $15 \pm 2\ \mu mol\ h^{-1}$ compared with both ZnS ($2.2 \pm 0.4\ \mu mol\ h^{-1}$) and plain CdS ($9 \pm 3\ \mu mol\ h^{-1}$) counterparts. The apparent quantum efficiency (AQE) at 420 nm is 1.4% for QDNWs (Supplementary Fig. 19), the value arising due to lack of efficient cocatalysts. Since the CdS component acts as light absorber, we normalized the evolution rates according to the total mass of CdS, which were determined using inductively coupled plasma mass spectrometry (ICP-MS)

analysis. In this way, the QDNWs achieved superior photocatalytic activities compared to prior cocatalyst-free CdS-based photocatalysts (Supplementary Table 1).

We sought to investigate further the underlying photophysical processes in quantum-dot-in-nanowire structures using transient absorption (TA) spectroscopy. To characterize the bandstructure of the 3CdS-4ZnS QDNWs, we first subtracted the absorption of Ag₂S tips and obtained the ground state transition energies from absorption spectra by fitting with Gaussian functions (see details in Supplementary Note 3, Supplementary Fig. 20, and Supplementary Table 2). Figure 4c shows the absorption spectrum and Gaussian-fitted first exciton transition ($1S_e{-}1S_h$) of the

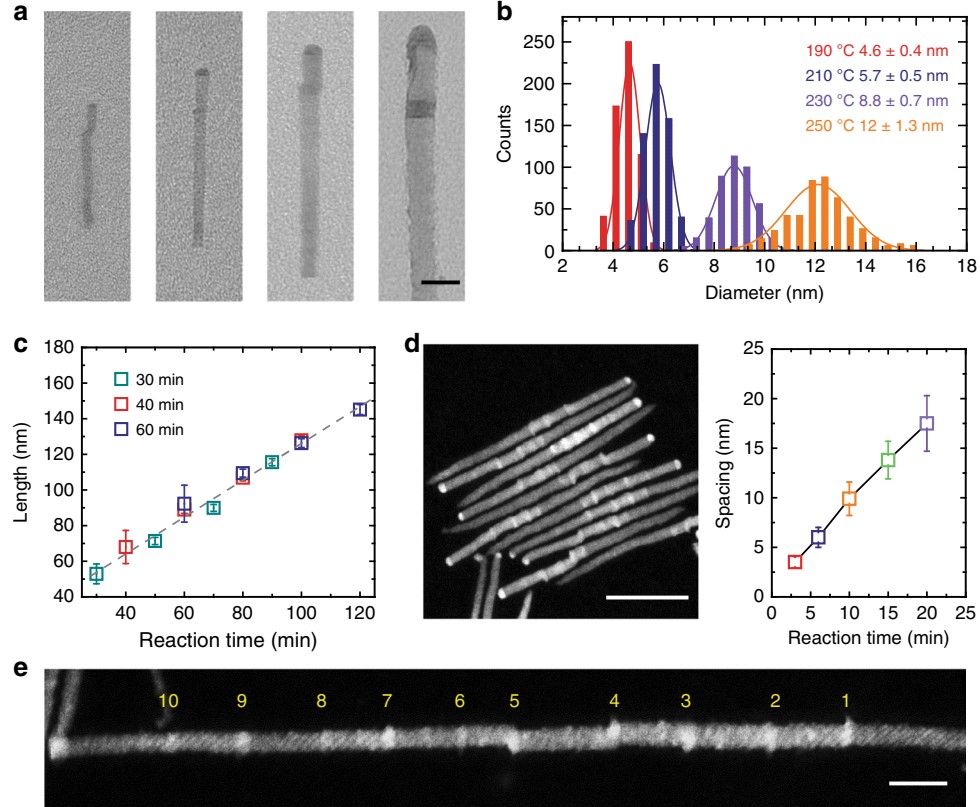

**Fig. 3** Modulation of geometric parameters in QDNWs. **a**, **b** Representative TEM images (**a**, from left to right) and the corresponding diameter distributions (**b**) of CdS–ZnS QDNWs grown at temperature of 190 °C, 210 °C, 230 °C, and 250 °C, respectively. **c** Time dependence of nanowire length for QDNWs with different initial ZnS growth time. Error bars in **c** correspond to standard deviation values of each newly grown ZnS segment. **d** (Left panel) Dark-field STEM image of QDNWs with modulated spacings of CdS QDs. (Right panel) The spacing of two adjacent CdS QDs (i.e., ZnS length) versus ZnS growth time. Error bars in **d** correspond to standard deviation values of each newly grown ZnS segment. **e** Representative STEM image of an ultralong QDNW with 10 CdS QDs (i.e., 21 segments in total). Scale bars are 20 nm, 50 nm, and 20 nm for **a**, **d**, and **e**, respectively

3CdS-4ZnS QDNWs. The bottom panel in Fig. 4c is a typical TA spectrum recorded at 50 ps following a 365 nm photoexcitation that selectively excites CdS. The ground state bleach (GSB) centered at 460 nm is attributed to the state filling of the $1S_e-1S_h$ excitonic transition. As the degeneracy of the valence band is much higher, the GSB is dominated by electrons in the conduction band[47]. We found that electrons in the $1S_e$ state of CdS–ZnS QDNWs depopulate faster than those of plain CdS QDs (Fig. 4d and Supplementary Fig. 21). We attribute the longer electron lifetime to a lower density of trap states in this selectively-passivated architecture, where the two (111) end facets of CdS QD account for around 42% of its total surface area. The photoluminescence (PL) spectra of QDNWs and plain QDs share similar features (Supplementary Fig. 22a), while the PLQY of QDNWs (11 ± 0.4%) is 4.5x higher than that of plain QDs (2.5 ± 0.2%), indicating improved passivation in the QDNWs (Supplementary Fig. 22b).

We further performed DFT calculations on the electronic structures of plain CdS QDs (Fig. 4e) and CdS QDs axially stacked in a nanowire (Fig. 4f and Supplementary Fig. 23), respectively, to understand the nature of trap state passivation. In the plain CdS QDs, trap states originate principally from (111) facets and strongly localize the photoexcited charges. The (111) facets of QDs stacked in a nanowire are well passivated, allowing charges to spread over the entire QD, a fact that facilitates charge transfer during photocatalytic processes. Overall, the DFT results indicate that epitaxy on the (111) facets balances surface passivation and charge transfer.

To extract efficiently the long-lived electrons in the QDNWs, we further decorated gold nanoparticles (NPs) on CdS QDs (Supplementary Figs. 24-26). Dark-field STEM images and EDS element mapping images of (3CdS/Au)-4ZnS confirm the site-selective nucleation of Au NPs on CdS QDs, which is attributed to the smaller lattice mismatch between CdS and Au than that between ZnS and Au (see details in Supplementary Note 4 and Supplementary Fig. 27). The amount of Au in (3CdS/Au)-4ZnS QDNWs, as quantified by ICP-MS, is about 4.5 wt% of the nanowires. TA decay kinetics show a fast decay of electron populations in CdS excited state after Au decoration, evidencing that the photoexcited electrons in CdS QDs are efficiently transferred to Au NPs. Such facilitated charge separation contributes to a substantially increased hydrogen evolution rate, reaching 34 ± 3 μmol h$^{-1}$ (20 mg photocatalyst with 3.2 mg CdS inside). The sample of 3CdS-4ZnS showed good photocatalytic stability with <8% degradation following 8 cycles (totally 32 h); while the (3CdS/Au)-4ZnS shows a modest photocatalytic stability with 25% degradation following 8 cycles. We observed ripening of Au NPs and the losses of Au NPs on parts of CdS segments (Supplementary Note 5 and Supplementary Fig. 28).

## Discussion

The PAEG strategy provides control over the composition, dimension, number, and spacing of the segments in QDNWs. The strategy may be extended to other colloidal QDNW systems by selecting judiciously the host catalyst, semiconductor material,

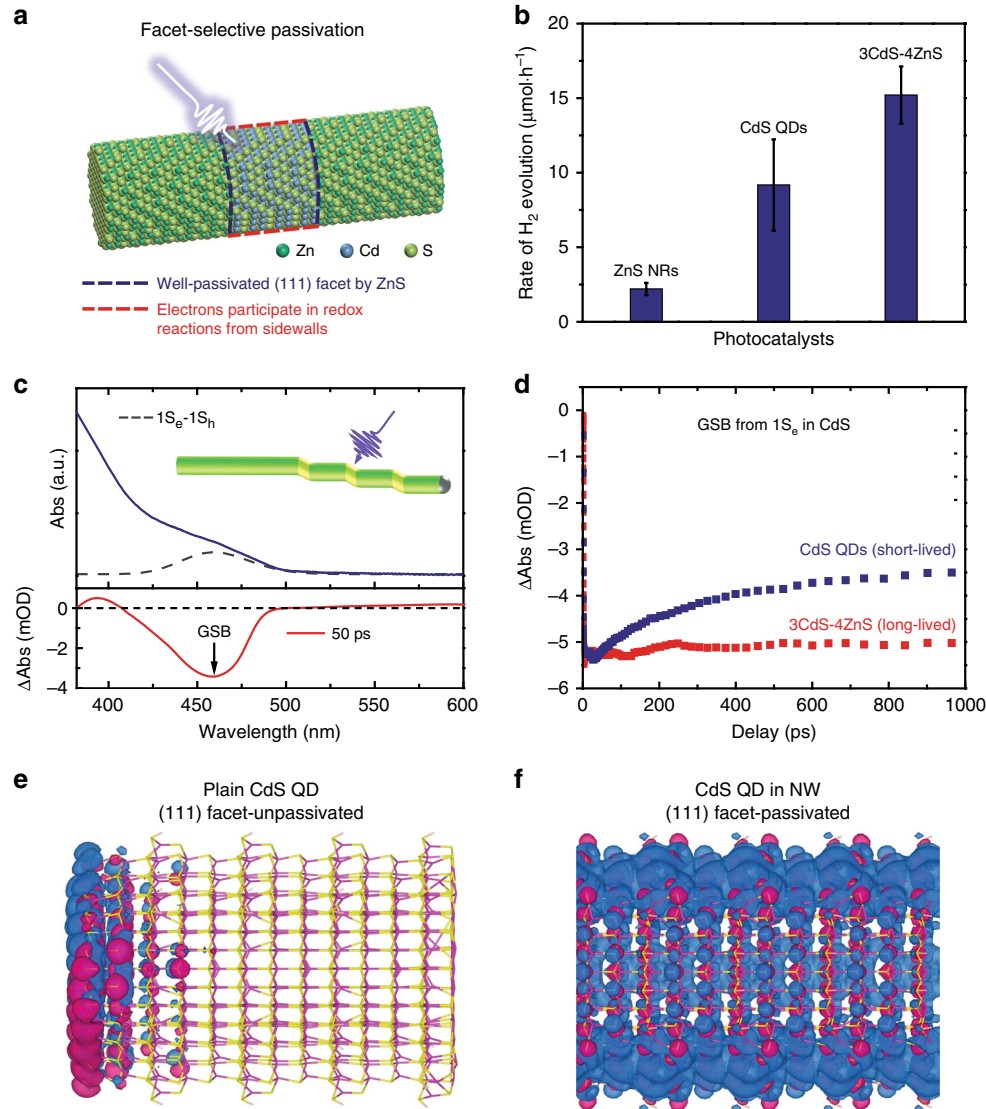

**Fig. 4** Facet-selective passivation for photocatalytic hydrogen production. **a** Schematic of selective passivation of CdS (111) facets by ZnS segments, allowing photoexcited electrons to participate in redox reactions from sidewalls. **b** Photocatalytic hydrogen evolution rates of different photocatalysts, including Ag$_2$S-tipped ZnS NRs, plain CdS QDs, and Ag$_2$S-tipped 3CdS-4ZnS QDNWs. **c** (Upper panel) Absorption spectrum (blue solid line) and Gaussian-fitted 1S$_e$–1S$_h$ exciton transition (grey dash line) of 3CdS-4ZnS QDNWs. (Bottom panel) A typical TA spectrum (red solid line) recorded at 50 ps after 365 nm pump. **d** TA decays for plain CdS QDs (blue) and 3CdS-4ZnS QDNWs (red) after 365 nm pump, monitored at the ground state bleach (GSB) in CdS component. **e**, **f** Spatial distributions of charge density of plain CdS QD (**e**) and CdS QD in a nanowire (**f**). Charges spread over the entire QD after (111) facet passivation

and precursor. QDNWs with fine-tuned structural parameters can be fabricated with the aid of programmed precursor injection using a syringe pump. The optical properties of QDNWs can be improved by shell passivation. Ultimately, QD-based applications in lasers[22], single-photon sources[48], and single-electron detectors[49] can be envisioned.

## Methods

**Synthesis of CdS–ZnS quantum-dots-in-nanowire.** First, the metal-diethyldithiocarbamate (dedc) precursors of Ag(dedc), Zn(dedc)$_2$, and Cd(dedc)$_2$ were prepared using a previously-reported method[31]. The ultrathin CdS–ZnS QDNWs were then synthesized via a catalyst-assisted method. Ag(dedc) (14.3 mg) and Zn(dedc)$_2$ (235.5 mg) were added into 10 mL 1-dodecanethiol in a three-neck flask and heated to 210 °C within 22 min. 10 mL oleic acid was then slowly injected after keeping the solution at 210 °C for 5 min. The reaction mixture color gradually turned turbid gray. After initial ZnS growth for a selected time (30-60 min), 40 mg Cd(dedc)$_2$ was added in situ into the reaction at a time interval of 20 min. An aliquot of products in each stage was taken and precipitated using ethanol for

analytical studies. The final QDNWs were collected and centrifuged. The products were washed twice with hexane and ethanol for further use. The synthetic procedures for other quantum-dot-in-nanowires with different structural parameters are similar to the above case, and are detailed in Supplementary Methods.

**Synthesis of CdS quantum dots.** CdS QDs with average diameter of 3.2 nm were synthesized in a similar reaction system to ensure the same crystalline and surface properties with the CdS segment in QDNWs. Typically, 250 mg Cd(dedc)$_2$ in 2 mL 1-dodecanethiol and 10 mL oleic acid was heated to 260 °C within 25 min and kept for 5 min before 10 mL oleic acid was injected into the hot solution. The reaction was kept for another 30 min and then cooled to room temperature. The resulting nanocrystals were cleaned using the procedure mentioned above.

**Synthesis of Ag$_2$S quantum dots.** 15 mg Ag(dedc) in 10 mL 1-dodecanethiol was heated to 140 °C within 15 min and kept for 10 min. The resulting nanocrystals were cleaned using the procedure mentioned above.

**Site-selective deposition of Au NPs on CdS of the QDNWs.** In the synthesis of ternary (3CdS/Au)-4ZnS QDNWs, 1 g AuCl$_3$ was first dissolved in 20 mL hexane

and 20 mL oleylamine, forming a clear yellow solution. Then 15 mg as-prepared 3CdS-4ZnS nanowires was well dispersed in 6 mL oleylamine in a three-neck flask and 150 μL HAuCl$_4$ solution was added. The reaction was heated slowly to 100 °C within 12 min under magnetic stirring and kept for 30 min. The resultant nanocrystals were collected and centrifuged at 8000 r.p.m. for 5 min. The products were further washed twice using hexane and ethanol.

**Photocatalytic water splitting experiments**. The hydrophobic nanomaterials were transferred from hydrophobic to hydrophilic media via a ligand exchange method in a two-phase solution[31,50]. Photocatalytic H$_2$ production was carried out in a Pyrex top-irradiation reaction vessel connected to a glass-closed gas circulation system. For each reaction, 20 mg catalyst was dispersed in an aqueous solution (100 mL) containing Na$_2$S (0.25 M) and Na$_2$SO$_3$ (0.35 M) as hole scavengers. The reaction solution was evacuated several times to remove air completely prior to visible-light irradiation under a 300 W Xe lamp (Newport Corporation) coupled with a UV-cutoff filter (>420 nm). The temperature of the reactant solution was maintained at 20 °C by a flow of cooling water during the reaction. Hydrogen was analyzed by a gas chromatograph (Agilent Technologies Corporation) equipped with a thermal conductivity detector. The final content of Cd in each photocatalyst was further quantified using ICP-MS analysis. The apparent quantum efficiency (AQE) was measured under the identical photocatalytic reactions by using 420 nm, 450 nm, 475 nm, 500 nm, and 520 nm band pass filters. The light intensity was calibrated using an irradiation meter. The AQE was calculated by equation (4):

$$AQE = 2N_H/N_P \times 100\%, \qquad (4)$$

where $N_H$ denotes the number of H$_2$ molecules and $N_P$ denotes the number of incident photons.

**Characterization**. The X-ray diffraction patterns (XRD) were measured using a Philips X'Pert Pro Super X-ray diffractometer equipped with graphite-monochromatized Cu Kα radiation ($\lambda = 1.5406$ Å). Nanocrystals dispersed in hexane were drop-cast on carbon-supported Cu grids for transmission electron microscopy (TEM) and high-resolution transmission electron microscopy (HRTEM) observations, which were performed using Hitachi H-7700 (Japan) and JEOL-2100F (Japan) with an acceleration voltage of 100 and 200 kV, respectively. Energy dispersive X-ray spectroscopy (EDS) with linescan and mapping modes and high-angle annular dark-field scanning transmission electron microscopy (HAADF-STEM) were carried out using Talos F200X electron microscope (FEI Inc.) operated at 200 kV. Inductively coupled plasma mass spectrometry (ICP-MS) was carried out on Optima 7300 DV to analyze Cd and Au content in each photocatalyst. UV-vis absorption spectra were collected on UV-2600 (Shimadzu Corporation, Japan) at room temperature with nanocrystals dispersed in toluene. Fourier transform infrared spectra (FTIR) were collected on Thermo Nicolet 6700. Photoluminescence spectra were collected using Fluorolog-3-Tou spectrometer (Jobin Yvon Inc.) at room temperature with nanocrystals dispersed in toluene. The absolute photoluminescence quantum yields (PLQY) were measured on an absolute PLQY spectrometer (c11347, Hamamatsu Inc.) with nanocrystals dispersed in toluene. All the size distributions of nanomaterials were analyzed by measuring more than 200 particles in TEM images using ImageJ software.

**Transient absorption measurements**. The 1030 nm fundamental (5 kHz) was produced using a Yb:KGW regenerative amplifier (Pharos, Light Conversion). A portion of this beam was sent through an optical parametric amplifier (Orpheus, Light Conversion) to generate the 3.4 eV photoexcitation pulse (pulse duration ~250 fs). Both the photoexcitation and fundamental were sent into an optical bench (Helios, Ultrafast). The fundamental, after passing through a delay stage, was focused into a sapphire crystal, generating the probe as a white light continuum. The frequency of the photoexcitation pulse was reduced to 2.5 kHz using a chopper. Both beams were then focused onto the sample, which was housed in a 1 mm cuvette. The probe was then detected by a CCD (Helios, Ultrafast). Samples were translated 1 mm/s during the measurement.

**Calculations on the relative intercalation energy**. We used ground state DFT with the Perdew-Burke-Ernzerhof (PBE)[51] GGA exchange correlation functional as implemented in the Vienna ab initio simulation package (VASP)[52] to perform structural optimization of the materials. All calculations allowed for spin polarization. We used plane wave energy cutoff of 520 eV and Gaussian smearing of 0.2 eV to converge the electronic problem. The Monkhorst-Pack k-points mesh of 900 atom$^{-1}$ and the force convergence criterion of 0.0005 eV per atom × N atoms in the unit cell were used as implemented in the MPRelaxSet class of the Pymatgen Python package[51–55]. 

The structure of the acanthite Ag$_2$S unit cell was taken from ref. [56]. To minimize the effect of image charge images interaction, the unit cell was replicated twice in x, y, and z directions resulting in a supercell of 96 atoms. In the intercalated supercells, one Ag$^+$ ion was replaced with a Zn$^{2+}$/Cd$^{2+}$ ion. Both the charged (+1) and neutral resulting supercells were considered.

**Calculations on the facet-selective passivation**. DFT calculations were performed using the Quickstep module of CP2K software[57], using a mixed Gaussians-plus-planewaves basis set. Godecker-Teter-Hutter pseudopotentials[58] were used along with PBE generalized gradients approximation exchange-correlation functional[51], molecule-optimized double-zeta basis set[59], and a 500 Ry grid cutoff.

The nanostructures of ~2.5 nm in diameter (~1000 atoms) were simulated in a (50 Å)$^3$ box. They were prepared starting from a zincblend CdS structure. All the surface atoms with 2 dangling bonds were passivated with an SH molecule, mimicking a thiol, or by creating an S-S dimer and converting these atoms to a configuration with a single dangling bond (XYZ structures attached). Similar results were obtained by terminating all surface Cd atoms having 2 dangling bonds with Cl, which is a computationally convenient model ligand that is electronically similar to oleic acid. Surface atoms with a single dangling bond were left unpassivated. Such surface structure underestimates the true surface passivation but is known to be sufficient to open the bandgap, i.e., eliminates the deep traps. Cd-rich (111) facet was left unpassivated, with the exception of several Cd vacancies to release the strain that arises from such unpassivated Cd adopting the planar sp$^2$ geometry. S-rich (111) facet was also left unpassivated as no L-type or X-type ligand can bind to electron-rich S dangling bonds. In the periodic case that is representative of a longer nanowire, traps on (111) facets are eliminated by the selective passivation.

## Data availability
The data that support the findings of this study are available on request from the corresponding authors (S.-H.Y. or E.H.S.).

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

## Acknowledgements

This work was supported by the National Natural Science Foundation of China (Grants 51732011, 21431006, 21761132008, 81788101, 11227901), the Foundation for Innovative Research Groups of the National Natural Science Foundation of China (Grant 21521001), Key Research Program of Frontier Sciences, CAS (Grant QYZDJ-SSW-SLH036), the National Basic Research Program of China (Grant 2014CB931800), the Users with Excellence and Scientific Research Grant of Hefei Science Center of CAS (2015HSC-UE007), Anhui Initiative in Quantum Information Technologies (Grant No. AHY050000), Ontario Research Fund – Research Excellence Program, and the Natural Sciences and Engineering Research Council (NSERC) of Canada. DFT calculations were performed on the IBM BlueGene Q supercomputer with support from the Southern Ontario Smart Computing Innovation Platform (SOSCIP).

## Author contributions

S-H.Y. and E.H.S. supervised the project. Y.L., T-T.Z., and F.F. conceived the idea. Y.L. carried out the experiments, analyzed the results, and wrote the paper. F.F. and T-T.Z. collected and analyzed the transient absorption spectra, and revised the paper. H.Z. helped to analyze the TA data. O.V. and M.A. performed DFT calculations. L.W. helped to analyze the crystal structure and growth mechanism. G-Q.L. and Y-X.P. helped with the photocatalysis experiments. All authors discussed the results and assisted during manuscript preparation.

## Additional information

**Competing interests:** The authors declare no competing interests.

