## [Peer Review File · Nature Communications]

Reviewers' comments:

Reviewer #1 (Remarks to the Author):

This is a good and well-presented study. The authors have devised a clever method for incorporating short QD-like CdS segments into ZnS nanowires grown by the SSS method. The 111 facets of the resulting CdS QDs thus become passivated by their epitaxial attachments to the ZnS segments. These passivated CdS QDs show modest to good enhancements in rates of photocatalytic water splitting. Given that the growth of nanowire heterostructures, including dots in wires, have been fairly extensively reported by vapor-based methods (as cited in manuscript), and even by solution-based methods (ref. 35), the originality and significance of this work are somewhat attenuated. I am not sure it has the importance normally associated with Nature Communications. I would welcome the editor to decide otherwise.

Specific issues:

1. Page 4, line 9 states: "However, these flask-based syntheses require chemical precursors to be added in a one-off approach, thereby inherently restricting dynamic control over precursor concentrations and resulting composition along the nanowire." This is deceptively worded. Technically, the solution-based method reported in ref. 35 does not use a flask, but it certainly does achieve control over precursor composition and produces heterostructured wires, as the authors do here. This achievement in ref. 35 should be explicitly acknowledged in the text.
2. Page 6, line 22 states: "Clearly, we can observe an increasing segment number." This sentence is unclear. Under what circumstances does one see increasing segment number? With the number of Cd additions? With an increasing length of the nanowires? What is meant here?

Reviewer #2 (Remarks to the Author):

The manuscript contributed by Li et al. reports an exciting class of colloidal structures of quantum dots in nanowires, which are synthesized through a pulsed axial epitaxy growth (PAEG) strategy developed by the authors. The success in synthesizing such complex semiconductor structures demonstrates a great potential in utilizing complex nanostructures for new applications or improvement of current applications. As demonstrated in the manuscript, CdS quantum dots are embedded in wider bandgap ZnS nanowires, significantly passivating the surface defects of CdS quantum dots to improve charge separation and transfer efficiency upon photoillumination. This leads to an enhancement of photocatalytic performance of the CdS-ZnS composite nanowires compared to the CdS quantum dots only. The results reported in this manuscript represents an excellent example to highlight the synergistic effect between different components in high-order and complex nanostructure. The unique design principles is so important for the community to create new materials with new properties and applications. The high quality of the results and manuscript makes it suitable for being published as soon as possible.

Although the photoemission should be avoided to improve photocatalytic activity, the reviewer is curious the difference of photoluminescence spectra between CdS-ZnS composite nanowires and CdS quantum dots. The corresponding spectra are suggested to be included in the revised manuscript or SI.

The CdS/ZnS interfaces are very clear in the HRTEM images shown in Figure 2g. Is there lattice mismatch between CdS and ZnS? What is responsible to accommodate the lattice mismatch? A review

paper overviewing the formation of binary interfaces (Natl. Sci. Rev. 2015, 2, 329), which is not cited here, might be helpful to look at the growth of CdS/ZnS interfaces.

Reviewer #3 (Remarks to the Author):

This manuscript (MS) reported a growth mode (pulsed axial epitaxy) to stack multiple CdS quantum dots into ZnS nanowires where solid Ag₂S was used as the host catalyst. Although Ag₂S-catalyzed growth of ZnS nanowires has been reported before (CrystEngComm, 2011, 13, 3515), the stacking of multiple CdS QDs in ZnS nanowire is still interesting and original. This flexible synthetic strategy allows precise modulation of quantum-dot size, number, spacing, and crystal phase. The resultant CdS-ZnS QDNWs exhibit higher photocatalytic hydrogen evolution rate than plain CdS quantum dots. I think it can be accepted by Nature Communications if the following concerns can be responded reasonably.

1) It can be seen from Fig. 21f that an obvious decline in the amount of H₂ occurred when (3CdS/Au)-4ZnS was used as photocatalyst for run 3 and run 4. Therefore, I have to doubt the recycle stability of these developed photocatalysts in photocatalytic performances. More cycle times are suggested to carry out, and more characterizations, such as TEM, XRD, FTIR are suggested to perform on the samples before and after photocatalytic reaction.

2) In the experiment of photocatalytic hydrogen production, the suspension containing catalyst needs to be stirred. It can be deduced that the CdS-ZnS QDNWs, especially the long nanowires, are easily broken under vigorous stirring, and this place of CdS QDs may be a fracture point, which is not conducive to its recycle stability.

3) As claimed by authors that "only the CdS component absorbs irradiation here". Why did the plain ZnS present photocatalytic activity for H₂ production (0.11 ± 0.02 mmol/h/gcat, Fig. 4b). If ZnS does have the photocatalytic activity for H₂ production in the experimental conditions (>420 nm), what is the reason (the band gap of ZnS is about 3.7 eV)? What role does ZnS play in this study, to anchor CdS QDs or construct heterostructure with CdS QDs?

4) Why was 3CdS-4ZnS QDNWs selected to perform the photocatalytic H₂ production? If it is the optimal samples for the highest amount of H₂ production, the comparison between 3CdS-4ZnS and others QDNWs is suggested to present.

5) In Table 1, the photocatalytic H₂ production performance of CdS-ZnS QDNWs was compared with other CdS-based photocatalysts reported in literature. In fact, many related references are not included. I suggest authors to further improve the table by consulting the literature. In addition, the H₂ production rate over the CdS-ZnS QDNWs is not so high (0.76 mmol h⁻¹ gcat⁻¹). If only the CdS component was counted, the H₂ production rate increased to 5 mmol h⁻¹ gCdS⁻¹, which is a high H₂ production rate. However, the unit of photocatalytic H₂ production rate should not be per gram of catalyst (simply because sufficient amount of photocatalyst absorbs all the photons and cannot increase the rate any further), unlike the case of conventional thermal catalysis. I suggest author to test the apparent quantum efficiency (AQE), which is an acceptable parameter for convenient comparison.

6) What is the driving force that drives Au nanoparticles (NPs) to selectively nucleate on CdS QDs? I suggest authors to present some explanation. The amount of Au NPs needs to be quantitative by characterization, such as ICP-MS.

“Pulsed axial epitaxy of colloidal quantum dots in nanowires enables facet-selective passivation”

Reviewer #1 (Remarks to the Author):

This is a good and well-presented study. The authors have devised a clever method for incorporating short QD-like CdS segments into ZnS nanowires grown by the SSS method. The 111 facets of the resulting CdS QDs thus become passivated by their epitaxial attachments to the ZnS segments. These passivated CdS QDs show modest to good enhancements in rates of photocatalytic water splitting. Given that the growth of nanowire heterostructures, including dots in wires, have been fairly extensively reported by vapor-based methods (as cited in manuscript), and even by solution-based methods (ref. 35), the originality and significance of this work are somewhat attenuated. I am not sure it has the importance normally associated with Nature Communications. I would welcome the editor to decide otherwise.

We thank the referee for these comments on the quality of the work and the novelty of the synthetic strategy. We now better explain that – comparing with prior vapor and solution based methods – the present advance provides, for the first time, a means to achieve the *highly controllable growth of quantum-confined heteronanostructures*. In detail:

- 1) In vapor-based and in certain solution-based (new ref. 36) methods, the composition of each segment in heteronanostructures can be well controlled; however, achieving small size within quantum confinement, which is critical to engineer high-efficiency photocatalysts with large surface-to-volume ratio and wide bandgap tunability, represents a need unmet prior to the present work.
- 2) Reducing nanostructure size can be achieved in colloidal synthesis, but there remains a dearth of reports of dynamically switching among heteronanostructure compositions.

Our pulsed axial epitaxy strategy overcomes prior limitations by utilizing catalyst tips to selectively grow different segments.

We have added the following statement to the main text:

“We hypothesized that trap state recombination on the (111) facets of QDs could be mitigated, yet carrier extraction from the sidewalls maintained, in this system, enhancing photocatalysis. However, these substrate-based methods produce thick nanowires with diameters of tens to hundreds of nanometers. Given that the large surface-to-volume ratio plays a key role in photocatalysts, and wide bandgap tunability offers new avenues for performance optimization, we sought strategies that would permit us to reduce the size of QDNW architectures” (page 3, line 20)

“Colloidal synthesis provides access to small-sized QDs and nanowires.” (page 4, line 6)

“Thus, we can achieve the dynamical switching of precursors in conventional flasks.” (page 5, line 1)

Specific issues:

1. Page 4, line 9 states: "However, these flask-based syntheses require chemical precursors to be added in a one-off approach, thereby inherently restricting dynamic control over precursor concentrations and resulting composition along the nanowire." This is deceptively worded. Technically, the solution-based method reported in ref. 35 does not use a flask, but it certainly does achieve control over precursor composition and produces heterostructured wires, as the authors do here. This achievement in ref. 35 should be explicitly acknowledged in the text.

We have revised as follows our portrayal of ref. 35 (new ref. 36):

“In particular, the in-situ growth of axial heteronanowires on a substrate has been achieved through a flow-based SSS method³⁶ – chemical precursors inside the microfluidic reactor were dynamically switched to grow the distinct segments. In colloidal syntheses, nanowires are instead suspended in solution, and precursors not readily removed as in substrate-based methods, a fact that restricts dynamic control over precursor concentrations and resulting composition along the nanowire.” (page 4, line 12)

2. Page 6, line 22 states: "Clearly, we can observe an increasing segment number." This sentence is unclear. Under what circumstances does one see increasing segment number? With the number of Cd additions? With an increasing length of the nanowires? What is meant here?

We have revised the manuscript to read:

“With increasing times of Cd addition, we observed an increasing number of CdS QDs in the nanowire, as seen in the contrast in the TEM images.” (page 7, line 5)

Reviewer #2 (Remarks to the Author):

The manuscript contributed by Li et al. reports an exciting class of colloidal structures of quantum dots in nanowires, which are synthesized through a pulsed axial epitaxy growth (PAEG) strategy developed by the authors. The success in synthesizing such complex semiconductor structures demonstrates a great potential in utilizing complex nanostructures for new applications or improvement of current applications. As demonstrated in the manuscript, CdS quantum dots are embedded in wider bandgap ZnS nanowires, significantly passivating the surface defects of CdS quantum dots to improve charge separation and transfer efficiency upon photoillumination. This leads to an enhancement of photocatalytic performance of the CdS-ZnS composite nanowires compared to the CdS quantum dots only. The results reported in this manuscript represents an excellent example to highlight the synergistic effect between different components in high-order and complex nanostructure. The unique design principles is so important for the community to create new materials with new properties and applications. The high quality of the results and manuscript makes it suitable for being published as soon as possible.

Although the photoemission should be avoided to improve photocatalytic activity, the reviewer is curious the difference of photoluminescence spectra between CdS-ZnS composite nanowires and CdS quantum dots. The corresponding spectra are suggested to be included in the revised manuscript or SI.

We thank the referee for these helpful comments. We now provide the photoluminescence (PL) spectra and absolute PL quantum yield (QY) in Fig. R1 (new Supplementary Fig. 22).

The PL spectra of QDNWs and plain QDs share similar features; while the PLQY of QDNWs (11%) is 4.5x higher than that of plain QDs (2.5%), indicating improved passivation in QDNWs.

We have added above discussion to the revised manuscript. (page 10, line 8)

Fig. R1 (new Supplementary Fig. 22). Photoluminescence spectra (a) and absolute photoluminescence quantum yield (PLQY) (b) of 3CdS-4ZnS QDNWs and plain CdS QDs.

The CdS/ZnS interfaces are very clear in the HRTEM images shown in Figure 2g. Is there lattice mismatch between CdS and ZnS? What is responsible to accommodate the lattice mismatch? A review paper overviewing the formation of binary interfaces (*Natl. Sci. Rev.* 2015, 2, 329), which is not cited here, might be helpful to look at the growth of CdS/ZnS interfaces.

We have now cited this review paper on heterogeneous nucleation (*Natl. Sci. Rev.* 2, 329 (2015)).

In the Stranski–Krastanov model, the overgrown materials form a 2D layer with a strained lattice to accommodate the substrate lattice - until the thickness of overgrown layer reaches its critical thickness. In our case, the lattice mismatch between the CdS (111) plane (d spacing of 0.33 nm) and the ZnS (111) plane (d spacing of 0.31 nm) is 6.45%. According to calculations published by Frank Glas (*Phys. Rev. B* 74, 121302 (2006)), the critical thickness of lattice-mismatched materials having a 7% mismatch on top of sub-10 nm thickness nanowires is much larger than the length of our CdS segments. Lattice strain can be therefore elastically released at the lateral surface.

We have added the discussion in the revised manuscript (page 7, line 22).

Reviewer #3 (Remarks to the Author):

This manuscript (MS) reported a growth mode (pulsed axial epitaxy) to stack multiple CdS quantum dots into ZnS nanowires where solid Ag₂S was used as the host catalyst. Although

Ag₂S-catalyzed growth of ZnS nanowires has been reported before (CrystEngComm, 2011, 13, 3515), the stacking of multiple CdS QDs in ZnS nanowire is still interesting and original. This flexible synthetic strategy allows precise modulation of quantum-dot size, number, spacing, and crystal phase. The resultant CdS-ZnS QDNWs exhibit higher photocatalytic hydrogen evolution rate than plain CdS quantum dots. I think it can be accepted by Nature Communications if the following concerns can be responded reasonably.

1) It can be seen from Fig. 21f that an obvious decline in the amount of H₂ occurred when (3CdS/Au)-4ZnS was used as photocatalyst for run 3 and run 4. Therefore, I have to doubt the recycle stability of these developed photocatalysts in photocatalytic performances. More cycle times are suggested to carry out, and more characterizations, such as TEM, XRD, FTIR are suggested to perform on the samples before and after photocatalytic reaction.

We now report stability studies based on 8 cycles (32 hours) for both the 3CdS-4ZnS and the (3CdS/Au)-4ZnS samples (Fig. R2 and new Supplementary Fig. 28f). The 3CdS-4ZnS sample shows good stability with <8% degradation following 8 cycles; while the (3CdS/Au)-4ZnS shows a more limited photocatalytic stability with ~25% degradation following 8 cycles.

Fig. R2 (new Supplementary Fig. 24f). Recycle stability in photocatalytic performances for 3CdS-4ZnS (yellow) and (3CdS/Au)-4ZnS (violet).

We further characterized the (3CdS/Au)-4ZnS sample before and after the stability test to understand the causes of performance degradation. There is no difference in the XRD patterns (Fig. R3a) and FTIR spectra (Fig. R3b) for both samples. However, we note in TEM images that the Au nanoparticles (NPs) become larger and inhomogeneously distributed following the stability tests (Fig. R3c-e).

Fig. R3 (new Supplementary Fig. 28). Characterizations of (3CdS/Au)-4ZnS QDNWs before and after photocatalytic stability test. (a) XRD patterns before and after reactions. (b) FTIR spectra before and after reactions. TEM images before (c) and after (d, e) reactions.

We attribute this to Ostwald ripening, corresponding to lowering of the total surface Gibbs free energy. A similar transformation from Au clusters to Au nanoparticles, and an analogous performance degradation, were also observed by Xu et al (Sci. Rep., 6, 22742 (2016)). Thus, the ripening of Au NPs and the losses of Au NPs on parts of CdS segments are sufficient to explain performance loss. The photocatalytic stability of QDNWs can be further improved by the coating using TiO₂ shell (Nat. Commun. 9, 1543 (2018)).

We have now added above discussions to the revised manuscript (page 11, line 9) and Supplementary Note 5.

2) In the experiment of photocatalytic hydrogen production, the suspension containing catalyst needs to be stirred. It can be deduced that the CdS-ZnS QDNWs, especially the long nanowires, are easily broken under vigorous stirring, and this place of CdS QDs may be a fracture point, which is not conducive to its recycle stability.

We have examined the shape changes of the nanowires after performance test (32 hours, Fig. R4a) or vigorous stirring for an extended period of time (24 hours, Fig. R4b). The nanowires remain substantially intact following stirring thanks to the epitaxial growth of CdS QDs on ZnS nanowires.

Fig. R4. (a) TEM image of 3ZnS-4CdS QDNWs after photocatalytic stability test (32 hours). (b) TEM image of ultralong QDNWs after vigorous stirring for 24 hours.

3) As claimed by authors that “only the CdS component absorbs irradiation here”. Why did the plain ZnS present photocatalytic activity for H₂ production (0.11 ± 0.02 mmol/h/g_{cat}, Fig. 4b). If ZnS does have the photocatalytic activity for H₂ production in the experimental conditions (>420 nm), what is the reason (the band gap of ZnS is about 3.7 eV)? What role does ZnS play in this study, to anchor CdS QDs or construct heterostructure with CdS QDs?

We have revised the manuscript to clarify that there are Ag₂S tips in all nanowires, and that these catalyze nanowire growth and contribute visible and near infrared photon absorption (new Supplementary Fig. 4a and Supplementary Fig. 20b). We provide the absorption spectra below for ease of reference (Fig. R5).

Fig. R5. UV-vis absorption spectra of Ag₂S-tipped ZnS NWs and plain Ag₂S QDs.

ZnS segments are part of heteronanowires: they passivate the defect-prone (111) facets of CdS QDs and enhance photocatalytic activity.

4) Why was 3CdS-4ZnS QDNWs selected to perform the photocatalytic H₂ production? If it is the optimal samples for the highest amount of H₂ production, the comparison between 3CdS-4ZnS and others QDNWs is suggested to present.

We chose 3CdS-4ZnS QDNWs because they show the highest H₂ production rate among the samples we tested. We now provide a photocatalytic performance comparison among all QDNWs (ZnS, 1CdS-2ZnS, and 2CdS-3ZnS) in Fig. R6 and new Supplementary Fig. 18.

Fig. R6 (new Supplementary Fig. 18). Photocatalytic H₂ production rate (a) and time-dependent H₂ productions (b) of different Ag₂S-tipped QDNWs (20 mg for each sample).

5) In Table 1, the photocatalytic H₂ production performance of CdS-ZnS QDNWs was compared with other CdS-based photocatalysts reported in literature. In fact, many related references are not included. I suggest authors to further improve the table by consulting the literature. In addition, the H₂ production rate over the CdS-ZnS QDNWs is not so high (0.76 mmol h⁻¹ g_{cat}⁻¹). If only the CdS component was counted, the H₂ production rate increased to 5 mmol h⁻¹ g_{CdS}⁻¹, which is a high H₂ production rate. However, the unit of photocatalytic H₂ production rate should not be per gram of catalyst (simply because sufficient amount of photocatalyst absorbs all the photons and cannot increase the rate any further), unlike the case of conventional thermal catalysis. I suggest author to test the apparent quantum efficiency (AQE), which is an acceptable parameter for convenient comparison.

We provide updated references summarized in Supplementary Table 1.

We now use the unit of μmol h⁻¹ to describe the H₂ production rate in the revised manuscript and supplementary information.

The AQE of our 3CdS-4ZnS QDNWs (Fig. R7 and new Supplementary Fig. 19) at 420 nm is 1.4%. We attribute the low efficiency to the fact that we did not incorporate an efficient cocatalyst (such as Pt and MoS₂). We have not yet succeeded in depositing these on our dots-in-nanowires and we are actively exploring solutions.

We have added the discussion in the revised manuscript (page 9, line 8).

Fig. R7 (new Supplementary Fig. 19). Apparent quantum efficiency (AQE, red dots) in photocatalytic H₂ production and the absorption spectrum (yellow dashed curve) of the 3CdS-4ZnS QDNWs.

6) What is the driving force that drives Au nanoparticles (NPs) to selectively nucleate on CdS QDs? I suggest authors to present some explanation. The amount of Au NPs needs to be quantitative by characterization, such as ICP-MS.

Lattice matching is responsible for the site-selective nucleation of Au NPs on CdS QDs. The d spacings of CdS (111) planes, ZnS (111) planes, and Au (111) planes are 0.335, 0.311, and 0.235 nm, respectively. The lattice mismatch between 3 layers of Au (111) plane and 2 layers of CdS (111) plane is 5.2%. In contrast, the lattice mismatch between the same layers of ZnS and Au is 13.3 %. The HRTEM image of (3CdS/Au)-4ZnS QDNWs, as shown in Fig. R8 (new Supplementary Fig. 27), accords well with our calculations. 10 layers of CdS (111) planes match well with 15 layers of Au (111) planes.

We have now added above discussion to the revised manuscript (page 11, line 1) and Supplementary Note 4.

Fig. R8 (new Supplementary Fig. 27). HETEM image of (3CdS/Au)-4ZnS QDNWs.

We have quantified the amount of Au in (3CdS/Au)-4ZnS QDNWs by ICP-MS, which is 4.5 wt% of the nanowires. We have now added this to the revised manuscript (page 11, line 3).

REVIEWERS' COMMENTS:

Reviewer #1 (Remarks to the Author):

I am generally satisfied with the changes made to the manuscript. On page 4, line 13 the authors claim that heteronanowire growth has been achieved by a flow-based SSS method, citing ref 36. That is incorrect. The method reported in ref 36 is a flow-based SLS (solution-liquid-solid) growth. This error requires correction.

Reviewer #2 (Remarks to the Author):

The revision clearly addresses the questions. It is now suitable for being published.

Reviewer #3 (Remarks to the Author):

The manuscript has been revised carefully and all comments I proposed have been responded reasonably. I think it can be accepted right now.

Reviewer #1 (Remarks to the Author):

I am generally satisfied with the changes made to the manuscript. On page 4, line 13 the authors claim that heteronanowire growth has been achieved by a flow-based SSS method, citing ref 36. That is incorrect. The method reported in ref 36 is a flow-based SLS (solution-liquid-solid) growth. This error requires correction.

We thank the reviewer for a constructive review process. We have corrected the error.

Reviewer #2 (Remarks to the Author):

The revision clearly addresses the questions. It is now suitable for being published.

We thank the reviewer for a constructive review process.

Reviewer #3 (Remarks to the Author):

The manuscript has been revised carefully and all comments I proposed have been responded reasonably. I think it can be accepted right now.

We thank the reviewer for a constructive review process.